# Effects of Different Doses of Caffeine Supplementation on Collegiate Male Volleyball Players’ Specific Performance and Skills: A Randomized, Double-Blind, Placebo-Controlled, Crossover Study

**DOI:** 10.3390/nu15184049

**Published:** 2023-09-19

**Authors:** Javad Nemati, Mohammad Hemmatinafar, Alireza Niknam, Mohammad Nikahd, Narjes Zeighami, Babak Imanian, Kousar Safari, Nima Jahaniboushehri, Katsuhiko Suzuki

**Affiliations:** 1Department of Sport Science, Faculty of Education and Psychology, Shiraz University, Shiraz 71345, Iran; 2Faculty of Sport Sciences, Waseda University, 2-579-15 Mikajima, Tokorozawa 359-1192, Japan

**Keywords:** caffeine doses, volleyball players, explosive power, agility, handgrip

## Abstract

Background: The improvement of performance and skill indices of volleyball players can affect their success rate. Therefore, the present study aimed to evaluate the effects of acute caffeine supplementation of varied doses on collegiate volleyball players’ specific performance and skills. Method: This research was a randomized, double-blind, crossover design study in which 15 male volleyball players aged 18 to 25 years participated voluntarily. Participants were randomly placed in three different conditions, including 3 mg of caffeine per kilogram of body weight (C3), 6 mg of caffeine per kilogram of body weight (C6), and a placebo (PLA) with a one-week wash-out period between exercise trials. The supplement was taken 60 min before each exercise session. Ball throwing, hand movement speed, agility, Sargent’s jump and handgrip, and attacking and serving skill tests were measured and analyzed to check the performance and skill of the volleyball players. Results: This study showed a significant increase in agility test in C6 compared with the PLA condition (*p* = 0.02). Additionally, there was a significant improvement in Sargent’s jump in C6 compared with PLA (*p* = 0.00) and C6 compared with the C3 condition (*p* = 0.00). Also, attacking skill significantly increased in C3 compared with PLA (*p* = 0.00) and C6 compared with the PLA condition (*p* = 0.00). In addition, there was a significant increase in serving skill for C6 compared with PLA (*p* = 0.00) and C3 (*p* = 0.00). However, there were no significant differences in hand movement speed (*p* = 0.06), left handgrip (*p* = 0.85), right handgrip (*p* = 0.47), or medicine ball throwing (*p* = 0.22) between the three conditions. Conclusions: In conclusion, the findings of the current study suggested that a higher dose of caffeine compared with a lower dose may be more effective in movements requiring lower-body explosive power and the ability to change direction. Also, according to the findings, it seems that caffeine can lead to the improvement of complex skills, such as serving and attacking in volleyball.

## 1. Introduction

With the participation of volleyball in the Olympic Games and the holding of attractive competitions such as the World League, Nations League, and domestic leagues of countries, the importance of players’ performance in this popular sport is being noticed more than before [1]. In addition, volleyball is an intermittent sport in which players perform high-intensity activities and skills with intervals of low-intensity activities. Hence, the optimal performance of aerobic and anaerobic energy systems is potentially important in the performance of volleyball players [2]. In this regard, it has been shown that bio-motor abilities such as vertical jump, upper-body explosive power, strength, agility, and speed of hand movements play an important role in the performance of volleyball players [2,3,4]. Also, previous studies have shown that the quality of technical skills, including serving, attacking, and blocking, can predict success in volleyball [5]. It should be noted that the ideal performance of athletes, including volleyball players, requires the coordination of different organs and systems of the body, especially the cardiovascular, neuromuscular, and endocrine systems. Therefore, strategies affecting the organism’s systems may also affect the technical and physical performance of the athletes. One of the most well-known strategies affecting different body systems is nutrition and nutritional supplements. Although the existing knowledge about how nutrition affects the performance of the body and athletes is still expanding, it is well known that nutrition can help improve sports performance through its acute and chronic effects [6,7]. Interestingly, some supplements with a quick effect on neuromuscular, metabolism, and cardiovascular function lead to the improvement of technical or physical performance of athletes [8,9]. Caffeine supplements are one of the most well-known nutritional agents, and they are widespread among athletes due to their acute effects on human performance [8].

Caffeine is rapidly distributed in the body and affects the central nervous, respiratory, cardiovascular, and musculoskeletal systems [10,11]. Caffeine blood level increases 15 min after consumption, and the peak of its increase is after 60 min [9]. In addition, caffeine is a strong moderator of central nervous system activity, which improves alertness or arousal and reduces sleepiness due to its antagonistic effect on adenosine receptors [12]. According to these effects of caffeine on the body, many studies have also shown that caffeine can have ergogenic effects in exercise training and competitions. Also, a meta-analysis study showed that the acute consumption of caffeine positively affected some components of team sports performance in female athletes [13]. However, several factors may play a role in the effectiveness of caffeine as an ergogenic aid, such as dose/dosage, volume of exercise training, timing of caffeine intake during the day, habitual caffeine consumption, and type of exercise [14,15]. Different doses of caffeine have been used in previous studies, and the results of these studies are mixed. For example, drinking a caffeinated beverage (with a dose equivalent to 3 mg of caffeine per kg body mass) has been shown to increase the jumping ability of male volleyball players [16]. Also, another study showed that adolescent basketball players’ jump performance improved after consuming a caffeine-containing energy drink (3 mg/kg body weight (BW)) [17]. Interestingly, a review study also reported that caffeine at lower doses (≤3 mg/kg) may also be ergogenic [18]. However, professional Costa Rican female volleyball players did not have any improvement in their physical performance after drinking an Energy Drink (including 76 mg caffeine), except for right handgrip strength [19]. Also, research data indicated that the combination of caffeine and placebo, or caffeine and carbohydrate (6 mg/kg caffeine capsules), did not improve the repeated sprint performance or agility of female basketball or volleyball players [20]. The data collected in one of the studies suggested that the amount of the supplement (50 mg of caffeine and averaged 1.39 mg/kg) used was ineffective in increasing female volleyball players’ vertical jump, agility, and repeated 30 m sprint ability [21]. However, another study reported that caffeine (5 mg/kg BW) could improve vertical jump performance in elite male volleyball players [22]. There is limited evidence on the effect of caffeine on specific skills in volleyball or other sports. A study has also shown that caffeine gum (with a dose of approximately 3 mg/kg BW) has ergogenic effects on attack accuracy, while no significant effect was observed on attack jump, vertical jump, block jump, sprint, or agility performance [23]. Another study also showed that caffeine gum (6.4 mg/kg BW) improves attack jump in female volleyball players [24]. The ergogenic effect of caffeine (3 mg/kg BW) on improving passing accuracy has also been shown in soccer players [25]. These studies have shown an ergogenic effect of caffeine on motor skills compared with a placebo but have not compared different doses of caffeine. Therefore, although more attention has been paid to the caffeine dose–response relationships regarding bio-motor capabilities, studies are still limited regarding motor skills, especially for volleyball.

Since there is limited evidence on the acute effect of different doses of caffeine on the specific skill performance of volleyball players, as well as inconsistent results about the effects of caffeine dose on the bio-motor abilities related to volleyball, it seems necessary to conduct more studies on this matter. Therefore, this study investigated the effect of different doses of caffeine (3 mg/kg BW vs. 6 mg/kg BW) on the skill and physical performance of male volleyball players.

## 2. Methodology

### 2.1. Participants

Fifteen male volleyball players with five years of volleyball experience voluntarily participated in this study. The demographic information of the participants is listed in Table 1. Participants had no known diseases or medical issues, no history of allergy to caffeine, and were not consuming any supplements or medications. All participants were fed the same breakfast containing 350–400 kcal (65% carbohydrates, 20% protein, and 15% fat) 2 h before the exercise test sessions. Participants were instructed to maintain their normal diet throughout the testing period, to avoid food and drink in the hour before testing, and to avoid strenuous exercise 24 h before each exercise test session. Participants were provided with a list of dietary sources of caffeine and asked to refrain from consuming these 24 h before each exercise test session. In addition, participants had access to drink water on a self-selected basis during the trials. This study was approved by the Ethics Committee of Shiraz University, Shiraz, Iran (SEP/14023/48/2441) and carried out by the Declaration of Helsinki.

### 2.2. Sample Size Calculation

The number of participants in this study was determined based on the study by Matsumura et al. [26], according to which caffeine ingestion led to a significant improvement in vertical jump compared with placebo (effect size  =  0.4). Using G*Power 3.1, considering the confidence interval of 95%, and the analysis power of 0.85, it was found that at least 11 participants were needed for this study. To ensure a sufficient sample size, 15 participants were selected for this study.

### 2.3. Study Design and Basic Measurements

This research was a randomized, double-blind, placebo-controlled, crossover design study. Before the implementation of the intervention, the participants were familiarized with the research process during a theoretical session. Also, the benefits and possible risks were explained to them. Then, participants completed the Physical Activity Readiness Questionnaire (PAR-Q) and the consent form for participation in the research (including full descriptions of the implementation method, benefits, risks, and possible complications). The height (Seca 217, Hamburg, Germany), weight, and fat percentage (InBody 270) of each participant were also measured in the familiarization session. Then, the participants were randomly divided into three different conditions of placebo (PLA), low-dose caffeine (3 mg/kg BW) (C3), and high-dose caffeine (6 mg/kg BW) (C6), with a crossover design, and performed three separate sessions of exercise testing with a one-week wash-out. The half-life of caffeine is reported to be approximately 6 h [27]. Therefore, to eliminate the effects of fatigue caused by exercise testing sessions and to ensure the complete clearance of caffeine and its metabolites from the body, a one-week washout was considered (Figure 1). In each session, participants performed the functional and skill tests after warm-up and consuming their relevant supplement (3 mg/kg BW (*n* = 5) or 6 mg/kg BW (*n* = 5) of caffeine supplement or placebo (*n* = 5)).

It should be noted that researchers were blinded to the conditions of each participant in the current study by an expert using the method of shuffling cards. In this method, the expert assigned the participants to three different conditions with codes 1, 2, and 3. The researchers did not know which condition was assigned to the PLA, C3, or C6. In addition, capsules of the same shape and size were used for each condition so that the participants were blind to each other’s condition. All participants were informed that they were taking a safe dose of caffeine.

### 2.4. Training Protocol

All participants were members of the same volleyball team, and their training regime was the same under the supervision of trainers. All subjects participated in the following training program: 4 training sessions of 90 min per week, including 10 min of warm-up, 20 min of physical training (core stability, jumping, agility, and quickness), 10 min of technical training, 20 min of tactical training, and 25 min of the training game, and at the end, there was cooling down for 5 min. Strength and power training occurred once per week as part of team training and consisted of a combination of plyometric (single-leg hops, drop jump, box jump, and squat jump: 3 sets × 12 repetitions for each) and resistance exercises (3–4 sets, 10–12 repetitions, and 75–80% of a maximum repetition). The type, intensity, load, and duration of the training program were similar for all participants.

### 2.5. Functional Test

The order of functional tests in each exercise test session was 1—handgrip strength, 2—Sargent’s jump, 3—upper-body explosive power (medicine ball throw), 4—attack skill, 5—service skill, 6—hand movement speed (plate tapping test), and 7—Illinois agility. Between each test, 5 min of active recovery was considered. To measure the maximum handgrip strength, a Pinch/Grip analyzer (MIE Medical Research Ltd., London, UK) was used. During the test, the elbow joint was in full extension and the shoulder was in 0-degree flexion (in standing position). The highest value that the dynamometer showed was recorded as the participants’ score. This test was performed for both hands separately and in two sets (with a 1 min interval between sets) [28]. The vertical jump test (Sargent’s jump) was used to assess the jumping ability of participants, and all assessments in the Sargent jump test were performed three times consecutively (1 min passive rest between them), with the best score recorded as the final result [29]. To measure the explosive power of the upper body, a medicine-ball-throwing test was performed, and each participant stood behind the throwing line and held the medicine ball under their chin. Then, without moving their legs, the participants leaned back and threw the medicine ball as far as they could towards the test field. The distance of the first contact of the ball with the ground and the throwing line was recorded as the score of each participant [30].

The Illinois test was used to measure the agility of the participants [31]. First, the participants lay prone behind the starting line (head towards the starting line and arms beside the body). Then, immediately after hearing the beep, they started to move and ran the test track with maximum effort. The test time was recorded using photoelectric sensors and timing gates (with a sensitivity of hundredths of a second) located at the beginning and end of the test path [31].

The Plate Tapping Test (Reaction Tap Test) was used to measure hand movement speed. In this test, two yellow disks of 20 cm in diameter were placed on a flat brown surface with their centers 60 cm apart. In between the two disks, a white rectangle (30 × 20 cm) was placed. The participants were asked to place their nonpreferred hand on the rectangle and move the preferred hand back and forth between the two yellow diss over the rectangle. Two taps were counted as one cycle, and the participants were asked to complete 25 cycles as fast as possible, and the time taken to complete the 25 cycles was counted using a stopwatch. Each participant performed the test twice, and the best performance was recorded [32].

### 2.6. Skill Test

To evaluate attacking proficiency, the line shot attack test was used [33]. This test was performed to evaluate the accuracy of linear attacks from the left/right side of the court. Each subject stood in their starting position 6.5 m from the net and 1.5 m from the left/right line (based on dominant hand) of the court. The setter was standing in an area with dimensions of 1.5 × 2 m. One side of the setter area was in the same line as the net, and the other side was three meters from the left/right side of the court. From the starting position, they underhand the ball with both hands in the setter area, where an expert setter performed an ideal setting to the attacking area. The expert setter used only an overhand set. The attacking area had dimensions of 2.5 × 1.5 m. One side of the attacking area was in line with the net, and the other side was in line with the left side of the court. In case a set was not ideal, it was canceled by the setter or the expert author and performed again. After the set, the attacker tried to score points using only line shot attacks in the target area, which consisted of four graded sections. The four graded point sections had the following dimensions: 4 = 0.5 × 0.5 m; 3 = 1 × 1 m; 2 = 1.5 × 1.5 m, and 1 = 2 × 2 m (Figure 2). The inner lines of every scoring area gave the highest number of points. The method of scoring evaluates a player’s ability to execute a line shot attack in a certain location of the opponent’s court in the left/right corner. The total score of ten trials was the final score. An excellent score was 40 points.

To determine the serve accuracy of the players at the overhand float serve, the standardized American Association for Health, Physical Education, and Recreation (AAHPER) serve test was used [34]. The test provides information on both the accuracy and the effectiveness of the server’s ability. The task of the participant was to complete 10 float serves from the serving zone (Figure 3), and they were instructed to serve hard and at the same time accurately, as they would do in an official game. Each subject was given two opportunities to make ten serves, and their highest total score was recorded.

### 2.7. Supplementation Protocol

In the current study, caffeine capsules were manufactured by the “Omid Mokmmel Salamat” Company, approved by the Food and Drug Organization of Iran, in doses of (3 or 6 mg/kg BW). Also, identical capsules with starch content were used for the placebo group. One hour before the test, the participants were given a supplement (3 mg/kg BW of caffeine or 6 mg/kg BW of caffeine or a placebo capsule). After that, the desired functional and skill factors were measured, and the second and third sessions were performed similarly (Figure 1). In addition, 48 h before the test, the subjects were advised to avoid caffeinated drinks and food. Food records were taken from the subjects on the day before the test to make sure that the subjects did not consume caffeinated foods on the day before the test.

### 2.8. Data Analysis

Data are expressed as the mean ± standard deviation (S.D.) and were analyzed using SPSS Windows software version 26.0. Repeated-measures analysis of variance (ANOVA) followed by Bonferroni’s post hoc testing was performed to determine statistical differences in performance and skill test results with characteristics over time between different time points [36]. EXCEL 2016 software was used to design graphs. The significance level was set to *p* < 0.05.

## 3. Results

Descriptive characteristics, including mean and standard deviation of the measured variables (medicine ball throwing, Sargent’s jump, strength of handgrip for the right and left hand, agility, hand movement speed, and attacking and serving skill), are reported in Table 2.

The one-way ANOVA repeated-measures analysis results show that the main effect on agility was significant (F = 4.77, *p* = 0.016, ⴄ^2^ = 0.25). Also, the results of the Bonferroni test indicate that agility in C6 increased significantly compared with PLA (MD = 0.64, *p* = 0.02, 95%CI [0.07–1.21]); however, there were no significant differences between C6 and C3 (MD = 0.1, *p* = 0.999, 95%CI [0.56–0.71]) and C3 and PLA (MD = 0.56, *p* = 0.09, 95%CI [0.07–1.21]) (Figure 4).

In addition, a significant effect was observed in Sargent jump height (F = 32.14, *p* = 0.00, ⴄ^2^ = 0.7). The results of the post hoc test demonstrate that Sargent jump height in C6 increased significantly compared with C3 (MD = 3.33, *p* = 0.000, 95%CI [1.93–4.72]) and PLA (MD = 4.26, *p* = 0.000, 95%CI [3.01–5.52]), while there was no significant difference between C3 and PLA conditions (MD = 0.93, *p* = 0.57, 95%CI [0.91–2.78]) (Figure 4). Furthermore, the analysis results show that the main effect on attacking skill was significant (F = 14.57, *p* = 0.000, ⴄ^2^ = 0.5). The results of the post hoc test show that C3 (MD = 2.26, *p* = 0.000, 95%CI [1.22–3.31]) and C6 (MD = 2.8, *p* = 0.001, 95%CI [1.11–4.48]) compared with PLA significantly increased attacking skill; however, there was no significant difference between C6 and C3 (MD = 0.53, *p* = 0.999, 95%CI [1.14–2.20]) (Figure 4).

Additionally, the results show that the main effect on serving skill was significant (F = 22.89, *p* = 0.000, ⴄ^2^ = 0.6). The post hoc test showed that C6 compared with C3 (MD = 2.46, *p* = 0.000, 95%CI [1.44–3.48]) and PLA (MD = 1.86, *p* = 0.000, 95%CI [0.87–2.85]) significantly increased serving skill, while there was no significant difference between C3 and PLA (MD = 0.6, *p* = 0.46, 95%CI [−1.68–0.48]) (Figure 4). however, the results demonstrate that there was no significant difference in hand movement speed (F = 3.73, *p* = 0.07, ⴄ^2^ = 0.21), left handgrip (F = 0.16, *p* = 0.85, ⴄ^2^ = 0.01), right handgrip (F = 0.60, *p* = 0.47, ⴄ^2^ = 0.04), and medicine ball throwing (F = 1.55, *p* = 0.22, ⴄ^2^ = 0.1) between the studied conditions (Figure 5).

## 4. Discussion

This study was conducted to investigate the effect of different doses of caffeine (3 and 6 mg/kg BW) on the performance of some physical (handgrip strength, vertical jump, agility, hand movement speed, and upper-body explosive power) and skill (attacking and serving) parameters of male volleyball players. The results of the study show that regardless of the dose, caffeine caused a significant improvement in the performance of the attack skill. However, agility showed a significant increase only after C6. Also, Sargent’s jumping performance and service skill in C6 increased significantly compared with C3 and PLA. In addition, there was no significant difference between C6, C3, and PLA in the performance of handgrip strength, upper-body explosive power, or hand movement speed. These findings suggest that a higher dose of caffeine compared with a lower dose may be more effective in movements requiring lower-body explosive power and the ability to change direction. Also, according to the findings, it seems that caffeine can lead to the improvement of complex skills, such as serving and attacking in volleyball.

Previous studies have also shown the positive effects of caffeine on agility performance [37,38,39], vertical jump [11,22,40,41,42,43], or special volleyball skills [23,24]. Several meta-analyses have also confirmed that caffeine can improve vertical jump [26,44,45,46] or agility performance [46,47]. However, some studies have also reported inconsistent results. For example, Lorino et al. [48] showed that caffeine (6 mg/kg BW) had no ergogenic effect on the agility performance of collegiate athletes. It should be noted that Lorino et al. used a different test compared with the present study to measure agility (5-0-5 proagility run test vs. Illinois test), which may partially explain the inconsistency of the results [48]. However, contrary to this hypothesis, Karampelas et al. reported that caffeine (5 mg/kg BW) did not have a significant effect on improving agility performance in the Illinois test [41]. Also, in the present study, only C6 (6 mg/kg BW) showed a significant effect on agility, which was more than caffeine used in the study of Karampelas et al. [41]. Also, the sample size of Karampelas et al.’s study was smaller compared with the present study, which may have influenced the observed results. In addition, Karampelas et al. [41] evaluated a smaller sample size compared with the present study (10 male soccer players vs. 15 male volleyball players). In this regard, a study with a similar sample size to the present study has shown that caffeine (6 mg/kg BW) can improve agility performance in the Illinois test [42]. Therefore, methodological differences, including the type of test to measure bio-motor and skill capabilities, sample size, caffeine dose, and training status of the participants, can explain the inconsistency between the findings of the present study and previous studies.

Some mechanisms (central and peripheral) explain the improvement of agility and vertical jump performance induced by caffeine in the present study. For example, it has been shown that caffeine (due to its effects on the CNS) leads to increased activation of motor units [49], which in turn improves the performance of athletes in bio-motor abilities, such as jumping and agility [26]. Caffeine’s antagonistic effect on adenosine receptors in the central and peripheral nervous systems leads to increased central drive and reduced effort/perceived pain [8,50]. In addition, caffeine exerts its influence on the central nervous system by eliciting the release of serotonin in the cerebral cortex, augmenting the performance of the sympathetic system, and diminishing the activity of inhibitory neurons [51,52]. Also, it has been suggested that caffeine improves anaerobic performance by increasing glycolytic flux (with stimulation of the sympathetic nervous and increased catecholamine release) and its direct effects on skeletal muscle (Ca^2+^ release from the sarcoplasmic reticulum, cross-bridge dynamics, and sodium–potassium–adenosine triphosphate pump activity) [8,53]. However, it has been reported that the release of Ca^2+^ is affected by high concentrations of caffeine, while the antagonistic effects of caffeine on adenosine receptors also appear in low concentrations of caffeine [49]. Since, in the present study, vertical jump performance and agility improved only after C6, the performance improvement may be more affected by peripheral mechanisms than the central mechanisms of caffeine. However, the plasma concentration of caffeine was not measured in the present study; so, it is difficult to distinguish between the effects of the peripheral and central mechanisms of caffeine on agility and vertical jump performance. Interestingly, some recent studies have confirmed that very low doses of caffeine and even placebo have ergogenic effects on vertical jump performance [26,39]. It is suggested that future studies pay more attention to the role of different mechanisms of caffeine on agility and vertical jump.

The findings have also shown that caffeine consumption improves performance in the volleyball attack test, regardless of the dose used. Also, volleyball serving performance increased after C6 compared with PLA or C3. Contrary to physical fitness parameters, there is very limited evidence about the effect of caffeine on volleyball-specific skill performance, especially service and attack accuracy. In addition, according to the knowledge of the authors, there is no study comparing the effect of different doses of caffeine and specific volleyball skills. However, in line with the findings of this study, it has been shown that caffeine gum consumption (at a dose of approximately 3 mg/kg body weight) improved attack accuracy [23]. In addition, another study also showed that chewing gum with high doses of caffeine (6.4 mg/kg of body weight) led to an increase in volleyball attack performance [24]. It should be noted that pure caffeine in capsule form was used in the current study; so, it seems that caffeine consumption regardless of the supplement form or dosage shows positive effects on improving accuracy and attack performance. In addition, two studies observed that consuming 6 mg/kg of caffeine improves successful shots and increases the accuracy of a tennis player’s serve [54,55].

Improved accuracy in volleyball attack and serve performance after caffeine consumption may be related to neurocognitive mechanisms. In this regard, it has been shown that acute consumption of 3 mg/kg BW of caffeine decreased the reaction time to a simple stimulus, decreased the time required to hit a fixed target, and increased the accuracy of hitting the target in a shooting game [56]. Interestingly, there is evidence that caffeine reduces the negative effects of alcohol on psychomotor performance and accuracy in reaction time tasks [57]. In addition, caffeine acutely increases the efficiency of the functioning of neural networks in the human cerebral cortex [58]. In turn, it can lead to increased performance in tasks that require attention through increasing alertness, concentration, and the ability to choose correctly among alternatives [59]. However, in the present study, information processing and central nervous system performance were not assessed; so, the improvement in attack and service test scores should not be solely attributed to these mechanisms. In addition, optimal performance in the attack test is influenced by jumping ability. Therefore, the improved results in the attack test may be also related to the underlying mechanisms of increasing jumping ability.

Also, this study showed that different doses of caffeine had no significant effect on upper-body explosive power, hand movement speed, or handgrip strength. However, the changes (relative percentage) compared with PLA showed that right handgrip strength (C3: 4.9% and C6:3.54%), upper-body explosive power (C3: 3.28% and C6: 3.38%), and hand movement speed (C3: 6.11% and C6: 6.02%) improved after caffeine supplementation. Consistent with these findings, A meta-analysis also showed that caffeine improved throwing performance with a small effect size, especially when the supplemental dose was ≤3 mg/kg BW [60]. Another study has also shown that caffeine has a small effect size on improving grip strength regardless of dosage [61]. Also, this study showed that the effect size is greater when the supplement is used in liquid form (Cohen’s d = 0.19) than in capsule form (Cohen’s d = 0.15). However, a study has also shown that caffeine consumption leads to a significant improvement in the handgrip strength and upper-body endurance of combat athletes [62]. In addition, genetic differences may also influence the effects of caffeine supplementation on strength parameters. For example, it has been shown that caffeine (4 mg/kg BW) negatively affected handgrip strength in competitive male athletes with the CC genotype of CYP1A2 [63]. Also, grip strength between caffeine and placebo did not show a significant difference in subjects with AA or AC genotype [63]. In the current study, the genotype of the participants was not investigated; so, genetic differences may have affected the results of the study. It is suggested that future studies pay more attention to the interaction of CYP1A2 genotype and caffeine ergogenic responses. It should be noted that the present study compared the effects of caffeine only with the placebo, while it has been shown that placebos have had ergogenic effects on performance compared with control conditions [64,65,66,67]. It is possible that due to the limitation of the present study design (lack of control conditions), the significant effects of caffeine on handgrip strength or upper-body explosive power are hidden compared with the control conditions. Hence, it is suggested that future studies consider the effect of placebo and caffeine on functional and skill parameters by considering the conditions/control group.

There were also limitations in this research. The main limitations of the current study were the lack of measurement of cognitive indicators, arousal, and other psychological variables affecting volleyball players’ function. Therefore, these cases should be corrected and controlled in future studies. Also, there was no control condition in the design of the present study, which may have hidden the effects of placebo or caffeine compared with the control. In addition, due to financial and time constraints, the authors of this study could not use the counterbalance design method to eliminate the order effect. Therefore, future studies should include the evaluation of these variables to expand on our findings.

## 5. Conclusions

In conclusion, the findings suggested that a higher dose of caffeine compared with a lower dose may be more effective in movements requiring lower-body explosive power and the ability to change direction. Also, according to the findings, it seems that caffeine can lead to the improvement of complex skills, such as serving and attacking in volleyball. Taken together, it is recommended to volleyball players that caffeine at a dose of 3–6 mg has ergogenic effects on jumping, ability to change direction, and accuracy of spikes and serves, without negative effects on grip strength, upper-body power, or hand movement speed.

## Figures and Tables

**Figure 1 nutrients-15-04049-f001:**
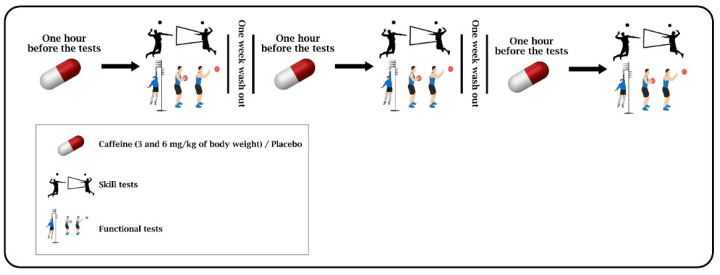
Study design and supplementation protocol.

**Figure 2 nutrients-15-04049-f002:**
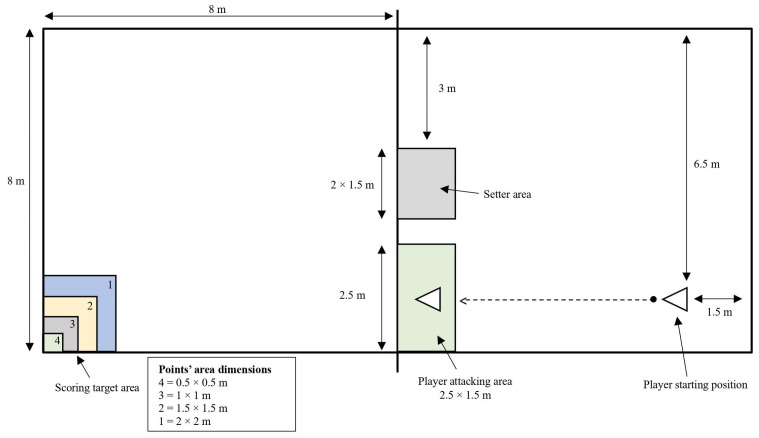
The line shot attack test instrument [33].

**Figure 3 nutrients-15-04049-f003:**
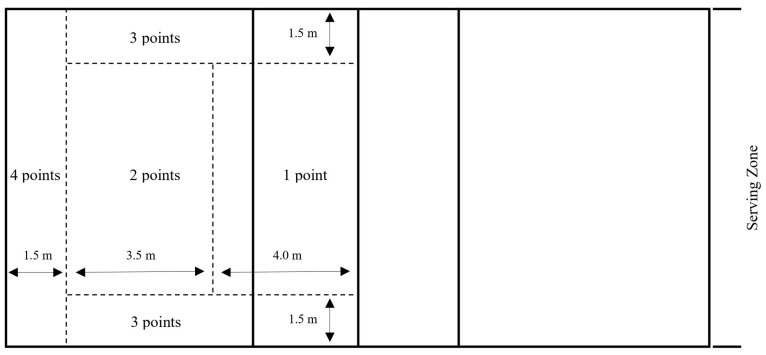
Scoring based on AAHPER serve test [35].

**Figure 4 nutrients-15-04049-f004:**
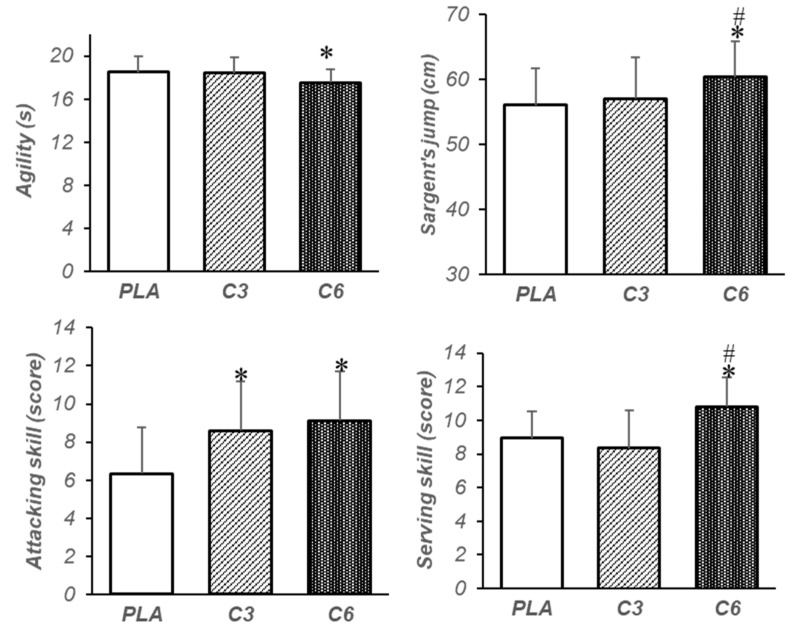
Mean and standard deviation (SD) of agility, Sargent’s jump, and attacking and serving skill for each condition. PLA: placebo, C3: 3 mg/kg BW of caffeine, C6: 6 mg/kg BW of caffeine, *: significant difference compared with placebo (*p* < 0.05), #: significant difference compared with C3 (*p* < 0.05).

**Figure 5 nutrients-15-04049-f005:**
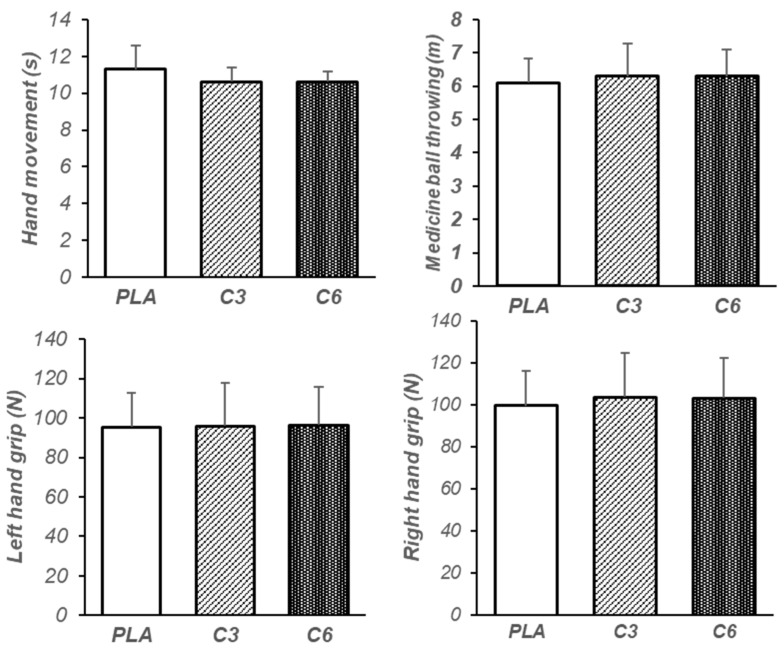
Mean and standard deviation (SD) of hand movement skill, medicine ball throwing, left handgrip, and right handgrip for each condition. PLA: placebo, C3: 3 mg/kg BW of caffeine, C6: 6 mg/kg BW of caffeine.

**Table 1 nutrients-15-04049-t001:** The demographic and characteristics of participants.

Characteristic	Mean ± SD (*n* = 15)
Age (years)	20.80 ± 1.00
Height (cm)	181.40 ± 4.30
Weight (kg)	70.22 ± 6.92
BMI (kg/m^2^)	21.00 ± 2.80
Fat mass (%)	19.00 ± 5.31

**Table 2 nutrients-15-04049-t002:** Mean and standard deviation (SD) of the tests results for each condition.

	PLA (*n* = 15)	C3 (*n* = 15)	C6 (*n* = 15)
	Mean ± SD	Mean ± SD	Mean ± SD
Medicine Ball (m)	6.10 ± 0.73	6.30 ± 0.97	6.30 ± 0.81
Sargent’s Jump (cm)	56.07 ± 5.57	57.00 ± 6.38	60.33 ± 5.51
Handgrip (Right) (N *)	99.60 ± 16.56	103.67 ± 20.69	103.13 ± 19.16
Handgrip (Left) (N *)	95.33 ± 17.56	95.80 ± 22.05	96.27 ± 19.55
Agility (s)	18.50 ± 2.21	18.42 ± 1.47	17.86 ± 1.48
Hand Movement Speed (s)	11.30 ± 1.29	10.61 ± 0.77	10.62 ± 0.58
Attacking (score)	6.33 ± 2.46	8.60 ± 2.61	9.13 ± 2.56
Serving (score)	8.93 ± 1.62	8.33 ± 2.25	10.80 ± 1.74

PLA: Placebo, C3: 3 mg/kg of body weight of caffeine, C6: 6 mg/kg of body weight of caffeine, *: N: Newton.

## Data Availability

Data sharing is applicable.

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
