# Peer review of "Effects of Different Doses of Caffeine Supplementation on Collegiate Male Volleyball Players’ Specific Performance and Skills: A Randomized, Double-Blind, Placebo-Controlled, Crossover Study"

_nutrients, 2023, doi:10.3390/nu15184049_

Round 1
Reviewer 1 Report
The methodology described in the provided text seems fairly detailed and structured, which is generally a good practice in research. However, there are a few potential areas where some improvements or clarifications could be made:
- Consistency in Units and Abbreviations:
- Inconsistent units and abbreviations can lead to confusion. For instance, "mg/kg BW" is used in some places while "mg per kilogram of body weight" is used in others. It's important to maintain consistency in your units and abbreviations throughout the methodology section.
- Randomization and Blinding:
- While the text mentions that the study was a randomized, double-blinded, placebo-controlled, crossover-design study, it would be beneficial to provide more information about the randomization process and how blinding was ensured. Details about how randomization was carried out and how blinding was maintained help ensure the study's validity.
- Informed Consent:
- While it's mentioned that participants completed a consent form, it's important to elaborate on the informed consent process. This should include details about how participants were informed about the study's purpose, procedures, potential benefits, risks, and their rights. This ensures that participants were adequately informed before giving their consent.
- Anthropometric Measurements:
- The text mentions that participants' anthropometric data was measured, but it doesn't specify which measurements were taken. Providing details about the specific anthropometric measurements (e.g., height, weight, body composition) would enhance the transparency of the study.
- Exercise Protocol Details:
- While the training protocol is described in general terms, adding more specific details about the skill, technical, and bodybuilding exercises would provide a clearer picture of the training regimen that the participants underwent.
- Test Protocols:
- The descriptions of the functional and skill tests are fairly detailed, but it's important to mention how the order of testing was counterbalanced across participants, especially in a crossover design. This helps prevent potential order effects.
- Supplementation Protocol:
- While the caffeine doses are mentioned, it would be helpful to specify how the participants were assigned to different supplementation groups (caffeine doses vs. placebo). This could include information about how the groups were balanced and whether any randomization was done specifically for the supplementation.
- Washout Period:
- The methodology mentions a one-week washout period between sessions. It's important to clarify how this duration was determined and why it was chosen. Justifying the washout period's length helps ensure that potential carryover effects are minimized.
- Pre-Test Restrictions:
- The text mentions that participants were advised to avoid caffeinated foods and drinks before the test, but it would be helpful to mention whether participants were given any other specific instructions regarding their activities, sleep, or diet in the days leading up to each session. These details can help control potential confounding variables.
- Participant Characteristics:
- Information about the participants' demographics (e.g., age, gender) and relevant characteristics (e.g., level of play, experience) is missing. Providing this information helps readers understand the sample population and its relevance to the study's objectives.
- Ethical Considerations:
- It's important to briefly discuss any ethical considerations that were taken into account, such as approval from an ethics committee or Institutional Review Board (IRB).
Remember that a well-structured methodology provides a clear and replicable framework for other researchers to understand and potentially replicate your study. Ensuring transparency and thoroughness in your methodology helps build credibility for your research findings.
Needs some revision
Reviewer 2 Report
Introduction
Enhance the overall flow between paragraphs to ensure a seamless progression of ideas. Additionally, refer to the manuscript for specific sections that might benefit from reorganization.
To offer a clearer overview of prior research findings, consider grouping similar outcomes and briefly summarizing these studies. This can be supported with references to specific sections of the manuscript where this synthesis has been done.
Explicitly state the research gap and outline the precise research question that the study aims to address. Refer to the manuscript for details on how the research question is formulated and highlighted.
Methods
Strengthen the method description by incorporating more specific details about the measurement techniques and procedures, enhancing the reader's understanding of how each test was conducted. Refer to the manuscript for sections that might benefit from elaboration.
Expand on the rationale behind the dosage selection for caffeine and placebo, highlighting the significance of using doses of 3 and 6 mg per kilogram of body weight. Provide more context regarding the potential impact of the supplementation on participants' physiological responses, referring to the corresponding section in the manuscript.
Clarify how the results were analyzed by providing a succinct overview of the statistical techniques used and why they were chosen. Consider referencing the manuscript for more detailed descriptions of the statistical analysis methods employed.
Results
Improve the clarity of result interpretation by providing more concise and focused descriptions of significant findings for each test rather than reiterating statistical details.
Enhance the graphical representation of the results by using clear labels for each condition (PLA, C3, C6) and indicating significant differences using appropriate markers. Ensure consistency in the labelling and presentation of figures throughout the section.
Discussion
While acknowledging contradictory findings, the discussion lacks a thorough analysis of potential reasons for discrepancies across studies. Expanding on factors such as participant demographics, exercise protocols, and caffeine dosages could enhance the section's depth.
Although the mechanistic explanations are insightful, the discussion could benefit from translating these mechanisms into practical implications for athletes, coaches, and sports professionals. Including actionable takeaways from the study's findings would enhance its real-world relevance.
While comparing two caffeine doses, the discussion fails to address the lack of a clear dose-response relationship. Exploring whether higher doses consistently lead to better outcomes or if an optimal range exists would provide a more nuanced interpretation.
While the discussion follows a logical structure, transitions between points could be smoother. Improving the flow between paragraphs and ensuring a coherent progression of ideas would enhance the readability and clarity of the discussion.
Conclusion
The conclusions drawn align with the results, but discussing potential limitations or addressing conflicting findings would strengthen the overall conclusions.
Moderate editing of English language required.
Round 2
Reviewer 2 Report
The authors did a good job in reviewing their manuscript.
Minor revisions.